# Dynamic Characteristics of Canopy and Vegetation Water Content during an Entire Maize Growing Season in Relation to Spectral-Based Indices

**Huailin Zhou** [1,2,3]**, Guangsheng Zhou** [1,2,4,]*****[ID]**, Xingyang Song** [1,3] **and Qijin He** [3,5]

1   State Key Laboratory of Severe Weather, Chinese Academy of Meteorological Sciences, Beijing 100081, China; zhouhl@nuist.edu.cn (H.Z.); songxy@cma.gov.cn (X.S.)
2   Joint Eco-Meteorological Laboratory of Chinese Academy of Meteorological Sciences and Zhengzhou University, Zhengzhou 450001, China
3   Hebei Gucheng Agricultural Meteorology National Observation and Research Station, Beijing 100081, China; heqijin@cau.edu.cn
4   Collaborative Innovation Center on Forecast Meteorological Disaster Warning and Assessment, Nanjing University of Information Science & Technology, Nanjing 210044, China
5   College of Resources and Environmental Sciences, China Agricultural University, Beijing 100193, China
*   Correspondence: zhougs@cma.gov.cn; Tel.: +86-10-68409148

**Abstract:** A variety of spectral vegetation indices (SVIs) have been constructed to monitor crop water stress. However, their abilities to reflect dynamic canopy water content (CWC) and vegetation water content (VWC) during the growing season have not been concurrently examined, and the underlying mechanisms remain unclear, especially in relation to soil drying. In this study, a field experiment was conducted and designed with various irrigation regimes applied during two consecutive growing seasons of maize. The results showed that CWC, VWC, and the SVIs exhibited obvious trends of first increasing and then decreasing within a growing season. In addition, VWC was allometrically related to CWC across the two growing seasons. A linear relationship between the five SVIs and CWC occurred within a certain CWC range ($0.01$–$0.41$ kg m$^{-2}$), while the relationship between these SVIs and VWC was nonlinear. Furthermore, the five SVIs indicated critical values for VWC, and these values were $1.12$ and $1.15$ kg m$^{-2}$ for the water index (WI) and normalized difference water index (NDWI), respectively; however, the normalized difference infrared index (NDII), normalized difference vegetation index (NDVI), and optimal soil-adjusted vegetation index (OSAVI) had the same critical value of $0.55$ kg m$^{-2}$. Therefore, in comparison to the NDII, NDVI, and OSAVI, the WI and NDWI better reflected the crop water content based on their sensitives to CWC and VWC. Moreover, CWC was the most important direct biotic driver of the dynamics of SVIs, while leaf area index (LAI) was the most important indirect biotic driver. VWC was a critical indirect regulator of WI, NDWI, NDII, and OSAVI dynamics, whereas vegetation dry mass (VDM) was the critical indirect regulator of NDVI dynamics. These findings may provide additional information for estimating agricultural drought and insights on the impact mechanism of soil water deficits on SVIs.

**Keywords:** canopy reflectance; crop water stress; spectral remote sensing; leaf area index; soil drying; structural indices; water indices



## 1. Introduction

Leaves are an essential component of plant canopy structure, and the water in leaves has a profound effect on photosynthesis, transpiration, and other physiological processes [1]. In addition, water storage in the stem provides a buffer between root water uptake and leaf transpiration [2]. Thus, plant water content is a key variable for inferring plant status and detecting plant adaptations to altered environments at multiple levels [3,4]. As a major abiotic stress, drought has a direct impact on physiological and biochemical processes, and the morphological structure of crop plants, ultimately contributing to yield loss or poor

crop quality [5–8]. Therefore, accurate real-time monitoring of crop water content is useful for enhancing our agricultural water management ability and improving the resource use efficiency of crops [9–11].

Several interrelated biophysical parameters can be used to quantify the amount of water in vegetation at different levels. Leaf water content ($C_w$) is defined as the total mass of water contained in a unit of leaf area or equal to the equivalent water thickness (EWT) [12,13]. Canopy water content (CWC), the total mass of water in all plant leaves per unit of ground area, is commonly applied to monitor plant growth [14–16]. At the same time, vegetation water content (VWC), the total amount of water in stems and leaves per unit of ground area, is a critical and indirect parameter for retrieving soil moisture from remote sensing data [17,18]. As CWC is sensitive to different water-deficit treatments, it is an optimum indicator for monitoring crop water stress [19]. Nonetheless, CWC accounts for only a fraction of the total plant water content, and the largest unknown for predicting VWC for a variety of vegetation types, except for grass species and very small plants, is stem water content [20,21]. In addition, research has demonstrated that VWC integrates the mechanisms involved in plant death: hydraulic failure and carbon depletion across organs and their interactions [22]. CWC has been shown to be linearly related to VWC for crops grown under normal conditions [21]; however, whether this linear relationship can be maintained under prolonged water-limited conditions remains unclear.

The amount of water present in vegetation can be estimated from remote sensors working in either optical or microwave spectral ranges [12,13,23]. Compared with those based on microwave spectral ranges, vegetation indices based on optical spectral ranges are more useful because of their higher spatial resolutions and better sensitivities to green vegetation, making them the baseline methods for CWC and VWC estimations [12,15,24]. Previous studies have suggested that CWC and VWC can be estimated by their regression relationship with various spectral vegetation indices (SVIs) derived from a combination of a reference band and/or water-absorption bands [13,25–28]. The most frequent SVIs applied to estimate CWC, VWC, and other related physiological variables, e.g., EWT and leaf area index (LAI), are the normalized difference vegetation index (NDVI), normalized difference infrared index (NDII), normalized difference water index (NDWI), water index (WI), and optimal soil-adjusted vegetation index (OSAVI) [29,30]. For instance, VWC appeared to have a linear relationship with the NDVI and NDWI [31,32], while its relationship was exponential with OSAVI [30]. In addition, the NDWI is more sensitive than the NDVI to leaf area and water content in closed canopies and tends to be saturated at a higher LAI [15]. Moreover, the relationship between the NDII and VWC was indirect and allometric, and the NDII was more related to CWC than to VWC [21]. Due to the limitations of destructive sampling methods or poor weather conditions during entire crop growing seasons, many studies are restricted to constructing spectral monitoring models for specific growth stages or a short growth period [5]. As the spectral characteristics of plants vary significantly across growth stages [33], the applicability of stage-specific or period-specific models to the entire growth period remains unknown. Specifically, how the SVIs mentioned above respond to CWC and VWC variations under soil water deficits has not been examined.

There are two different concepts related to plant water stress: instantaneous water stress and long-term water stress [34]. Instantaneous water stress generally lowers leaf water content, while long-term stress significantly affects crop physical parameters, such as LAI, canopy height, and biomass [35–38]. During a prolonged drought event, the effects of a water deficit on plants are continuous and progressive. However, due to the strong association between biomass accumulation and plant water content dynamics, the sensitivity of the SVIs to changes in canopy biomass and water content characteristics is unclear. In this study, we used an experimental approach to examine the relative sensitivity of five SVIs (WI, NDWI, NDII, NDVI, and OSAVI) for plant water stress and explore the underlying mechanism of the water stress effect on canopy reflectance characteristics, especially through the dynamic variations in canopy biomass and water content. Specifically, the objectives were to address the following questions: (i) What are the dynamic characteristics of

CWC, VWC, and SVIs during the entire growing season? (ii) How does VWC vary with CWC under soil drying during the growing season? (iii) Do the SVIs exhibit a significant difference in their abilities to determine CWC and VWC? (iv) What are the direct and indirect pathways by which crop biomass and water content influence spectral indices?

## 2. Materials and Methods

### 2.1. Study Site Description

This study was based on a maize field experiment conducted at the Hebei Gucheng Agricultural Meteorology National Observation and Research Station (39°08′ N, 115°40′ E and 15.2 m a.s.l.), Hebei Province, China. The region is in the central part of the North China Plain (NCP) and is characterized by a warm continental climate and temperate monsoon climate. The 30-year (1981–2010) average annual temperature, annual sunshine amount, and annual active accumulated temperature ($\geq$10 °C) were 12.2 °C, 2264 h, and 4910 °C, respectively. The mean annual precipitation is 515.5 mm and mainly occurs in summer (approximately 70%), even with a coefficient of variation of 62.9% [39]. The typical soil type is sandy loam, with the total nitrogen at 0.98 g kg$^{-1}$, total phosphorus at 1.02 g kg$^{-1}$, and total potassium at 17.26 g kg$^{-1}$ within 50 cm soil profiles. For a depth of 50 cm soil profile, the mean pH and soil bulk density are 8.19 and 1.37 g cm$^{-3}$, respectively [40]. Meanwhile, the field capacity and wilting coefficient are 22.7% and 5.0%, respectively [41]. Specifically, the field capacities of 0–10 cm, 10–20 cm, 20–30 cm, 30–40 cm, and 40–50 cm were 22.2%, 21.2%, 23.3%, 22.3%, and 24.7%. The farming practices in this region are double cropping with two crop harvests in a year (a wheat–maize rotation). The principal cultivated grain crops are spring wheat (*Triticum aestivum* L.) and summer maize (*Zea mays* L.).

### 2.2. Field Experimental Design and Crop Management

The maize variety used for this study was a drought-tolerant cultivar named zhengdan 958, which is the most popular maize variety in NCP since 2003. A randomized complete block design was employed in the field experiment. Each plot was 4 m long and 2 m wide, with an area of 8.0 m$^2$ per plot. There was a large electric-powered waterproof shelter over the experimental plots, which could block the rainfall when it was rainy. At other times, the waterproof shelter was moved away and the plots were exposed to the ambient conditions. A concrete wall (3 m deep) was constructed to prevent soil water exchange horizontally among plots. The soil moisture within a depth of 1.0 m was measured in each plot at one month before seeding. After calculation, each plot was irrigated and was maintained at the same soil water moisture. During the two maize growing seasons of 2013 and 2014, we sowed the maize seeds on 27 June 2013, and 24 June 2014, respectively. The plant density was set to 52 plants per plot (65,000 plants hm$^{-2}$). Prior to irrigational treatments, some irrigation was applied in each plot to improve seedlings emergence. In this study, five irrigation amounts were conducted at July 24, 2013 (seven-leaf stage) and July 2, 2014 (three-leaf stage) during the 2013 and 2014 growing seasons. The local mean precipitation in July during 1981–2010 was 150 mm, which was the basic reference for irrigation amount in this study. In 2013, five irrigation amounts (named T1–T5 treatments) were 80, 60, 40, 25, and 15 mm, respectively, equivalent to 53.3%, 40%, 26.7%, 16.7%, and 10% of the local mean precipitation in July (150 mm). In 2014, another five irrigation treatments (named W1–W5 treatments) were 150 mm 120 mm, 90 mm, 60 mm, and 30 mm, respectively, equivalent to 100%, 80%, 60%, 40%, and 20% of the local mean precipitation in July (150 mm) [42]. Each treatment was designed with three replicate plots. After irrigation, no additional irrigation water was applied, and precipitation was blocked during the remaining duration of the growing seasons. The controlled-release fertilizer diammonium phosphate (CRP) was applied in all treatments with a rate of 320 kg hm$^{-2}$. The harvest dates were on 8 October 2013, and 9 October 2014, respectively.

*2.3. Measurement*

2.3.1. Soil Water Content

The oven-drying method was applied to determine the soil water content with 7–14 days intervals throughout the entire growing season. Due to over 95% of the maize root biomass being grown within a depth of 30 cm soil layers [43], we set the sampling depth to 50 cm. We collected the soil samples at every 10 cm soil layer. Collected samples were dried in a ventilated oven at 105 °C until reaching a constant weight. We determined the mean soil moisture for each treatment from three different sampling sites in the middle area of each plot. During the 2013–2014 growing seasons, seven measurements were performed in each year. Available soil water content (ASWC, %) was calculated according to the following equations [44,45]:

$$ASWC = \frac{SWC - WP}{FC - WP} \times 100\% \tag{1}$$

where $SWC$ (%) is the measured soil water content as a percent of the dry soil weight. $FC$ (%) and $WP$ (%) are the field capacity and the wilting point, respectively.

2.3.2. Canopy Biomass and Water Content Characteristics

To obtain the vegetation water status and other canopy structure characteristics, we randomly selected three healthy maize plants from each treatment and harvested them during each survey. The sampling interval was identical to that of the soil water content measurement. During each measurement, all aboveground parts were harvested and were weighed timely. LAI was measured concurrently with aboveground biomass measurements. The maximum length ($L$, cm) and width ($W$, cm) of each blade were measured to calculate the plant leaf area according to a method suggested by a previous study [46]. The plant leaf area and leaf area index (LAI, $m^2 \, m^{-2}$) was determined as follows:

$$LA = \sum_i^n (0.75 \times L_i \times W_i) \tag{2}$$

$$LAI = \sum_j^k (LA_j)/k \times d/10000 \tag{3}$$

where $LA$ is plant leaf area ($cm^2$), $n$ is the total number of leaves per maize plant, $i$ is the $i$th leaf of maize plant, $k$ is the number of repetitions, and $d$ is the plant density in the plot (plant $m^{-2}$).

To obtain plant water content and dry biomass, all fresh plant parts were dried in an oven at 80 °C for more than 24 h until their weights were constant. The water content and dry biomass contained in leaves and aboveground organs were calculated by using the following formulas:

$$CWC = \sum_j^k (LFW_j - LDM_j)/k \times d/1000 \tag{4}$$

$$VWC = \sum_j^k (LFW_j - LDM_j + SFW_j - SDM_j)/k \times d/1000 \tag{5}$$

$$VDM = \sum_j^k (LDM_j + SDM_j)/k \times d/1000 \tag{6}$$

where $CWC$ (kg $m^{-2}$) is the water content in all green leaves per $m^2$, $LFW$ is the fresh weight of all green leaves (g $plant^{-1}$), LDM is the dry mass of green leaves (g $plant^{-1}$), $k$ is the number of repetitions, $d$ is the plant density of each plot (plant $m^{-2}$), VWC (kg $m^{-2}$) is water content in all leaves and stems per $m^2$, SFW is the stem fresh weight (g $plant^{-1}$), and SDM is the stem dry mass (g $plant^{-1}$). VDM (kg $m^{-2}$) is the dry mass of all leaves and stems per $m^2$.

### 2.3.3. Canopy Spectral Reflectance

Canopy spectral reflectance was collected using an ASD FieldSpec 3 Spectroradiometer (Analytical Spectral Devices, Inc., Boulder, CO, USA), which was calibrated using a 0.4 m × 0.4 m BaSO$_4$ calibration panel. The wavelength range was 350–2500 nm, with sampling intervals of 1.38 nm for 350–1050 nm and 2.0 nm for 1000–2500 nm. Field canopy spectral measurements were concurrent with those of other crop variables and soil water content measurements. All canopy spectral measurements were taken under clear sky conditions between 11:30 and 14:00 h to ensure maximum solar intensity when the sun was shining directly on the plants, as conducted in previous similar studies [47,48]. To minimize the noise of the soil background, the spectrometer was positioned at approximately 1.0 m above the canopy, and at a nadir angle with a field of view (FOV) of 25° so that it only viewed the plant canopy. A reflectance spectrum for each plot was determined as the average of the 15 spectra, and a mean spectrum per treatment was calculated as the average of the three plot replications. In addition, calibration measurements were made every 10 min. The spectral indices used in this study were obtained from the optical reflectance measurements and were presented in Table 1. The SVIs were divided into two groups: (1) the structural/greenness indices related to the canopy structure and biomass; (2) the water indices related to the water content of the vegetation.

**Table 1.** Spectral vegetation indices used to estimate plant water stress in this study.

| Names | Index | Formula | Reference |
|---|---|---|---|
| Water Indices | | | |
| Water Index | WI | $\frac{R900}{R970}$ | [49] |
| Normalized Difference Water Index | NDWI | $\frac{R860-R1240}{R860+R1240}$ | [10] |
| Normalized Difference Infrared Index | NDII | $\frac{R850-R1650}{R850+R1650}$ | [50] |
| Structural Indices | | | |
| Normalized Difference Vegetation Index | NDVI | $\frac{R800-R670}{R800+R670}$ | [51] |
| Optimized Soil Adjusted Vegetation Index | OSAVI | $(1+0.16) \times \frac{R800-R670}{R800+R670+0.16}$ | [52] |

*R* denotes the reflectance value at the indicated wavelength in nm.

### 2.3.4. Meteorological Conditions

The meteorological data during the maize growing seasons of 2013 and 2014 were collected by an on-site automated weather station (Figure S1), and the data included air temperature (*T*, °C), precipitation, photosynthetically active radiation (*PAR*, MJ m$^{-2}$), and air relative humidity (*RH*, %).

### 2.4. Statistical Analysis

Repeated measures analysis of variance (RM-ANOVA) was applied to evaluate the effects of the soil drying on CWC, VWC, and the SVIs among different treatments. The least significant difference (LSD) test was employed to distinguish the differences among treatments with Duncan's test. The relationship between CWC and the SVIs was represented by simple linear regression. To explore the connection between CWC and the SVIs, a piecewise linear regression was conducted. In addition, correlation analysis was conducted to determine the relationship between canopy characteristics and the SVIs. A structural equation model (SEM) was employed to analyze causal and interactive pathways influencing canopy or vegetation reflectance during soil drying by AMOS 21.0 (Amos Development Co., Greene, ME, USA). Base models were established from prior knowledge about the possible effects of explanatory variables on SVIs. Differences or correlations were significant at a level of $p < 0.05$.

## 3. Results

### 3.1. Dynamic Variations in CWC and VWC during Soil Drying in the Growing Season

Progressive soil drying significantly decreased CWC and VWC based on RM-ANOVA (Figure 1). The dynamic characteristics of CWC and VWC shared a similar trend of first increasing and then decreasing in both growing seasons (Figure 1a,b,d,e). The CWC and VWC values of the T1–T5 treatments concurrently reached maximum values at 70 DAS. In contrast, the date when CWC and VWC approached their maximum values was delayed with the irrigation reduction for the W1–W5 treatments in the 2014 growing season. In addition, the ratio of CWC to VWC exhibited a decreasing trend, and the difference among the T1–T5 treatments in 2013 was less than that among the W1–W5 treatments in 2014 (Figure 1c,f).

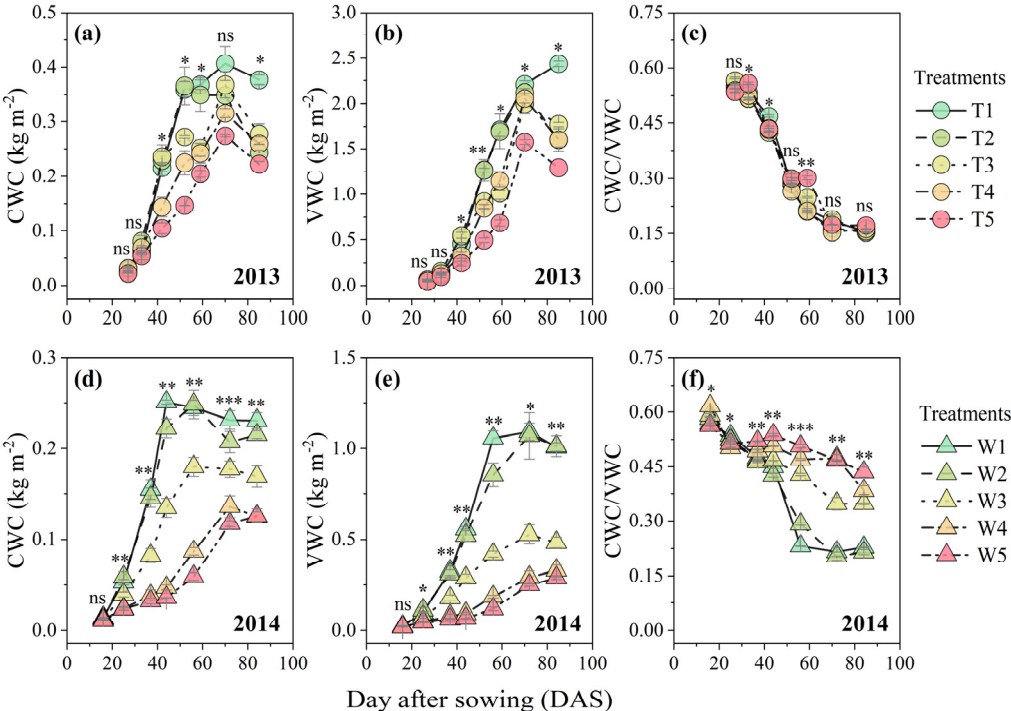

**Figure 1.** Changes in canopy water content (CWC), vegetation water content (VWC), and the ratio of CWC to VWC during soil drying under different irrigation treatments in the 2013 (**a–c**) and 2014 (**d–f**) growing seasons. (**a**) CWC in 2013, (**b**) VWC in 2013, (**c**) CWC/VWC in 2013; (**d**) CWC in 2014, (**e**) VWC in 2014, (**f**) CWC/VWC in 2014. The "***", "**", and "*" denote significant differences among treatments based on the analysis of variance (ANOVA) at $p < 0.001$, $p < 0.01$, and $p < 0.05$ levels, respectively, while "ns" indicates no significant differences. Error bars denote mean $\pm$ SE.

### 3.2. Relationship between CWC and VWC, and Their Connection with VDM during Soil Drying

VWC shared a significant power relationship with CWC during soil drying in the growing season (Figure 2). In 2013, VWC had a positive power function with CWC based on the pooled data. In 2014, the connection between VWC and CWC was also indicated by a power function (with a slightly higher power value of 1.62). Furthermore, CWC increased with increasing VDM before reaching 0.17 kg m$^{-2}$ and 0.12 kg m$^{-2}$ for the 2013 and 2014 growing seasons, respectively, and CWC remained relatively stable thereafter (Figure 3a,b). At the same time, VWC exhibited a significant correlation with VDM in both growing seasons (Figure 3c,d). In general, in response to water stress, the growth characteristics of the maize plants were similar even in different irrigation treatments and growth stages. Therefore, the spectral characteristics were analyzed with the pooled data collected from the two growing seasons.

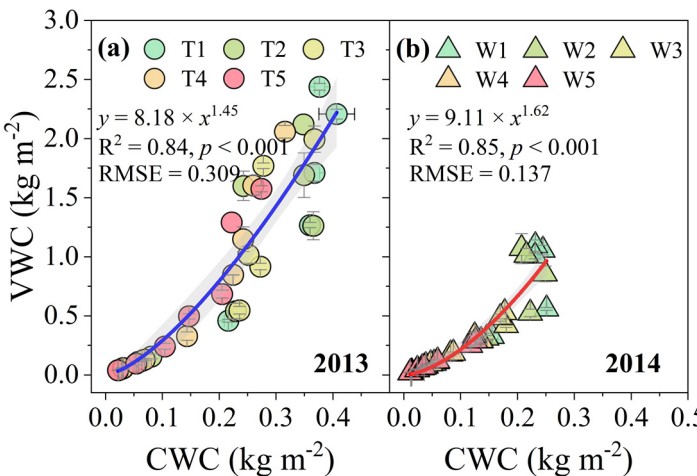

**Figure 2.** Connection between canopy water content (CWC) and vegetation water content (VWC) during soil drying under different irrigation treatments in the 2013 (**a**) and 2014 (**b**) growing seasons. Nonlinear regression lines show the 95% confidence intervals. Error bars denote mean ± SE.

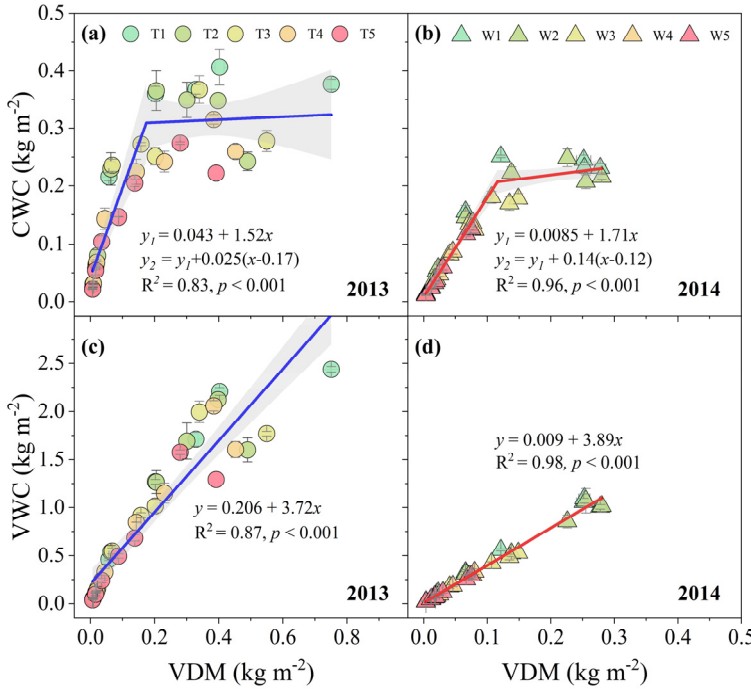

**Figure 3.** Relationship between canopy water content (CWC), vegetation water content (VWC), and vegetation dry mass (VDM) during soil drying under different irrigation treatments in the 2013 (**a,c**) and 2014 (**b,d**) growing seasons. Linear and nonlinear regression lines show the 95% confidence intervals. Error bars represent mean ± SE.

### 3.3. Dynamic Variations in the SVIs during Soil Drying in the Growing Season

The temporal variability in the SVIs also showed an obvious trend of first increasing and then decreasing during the 2013–2014 growing season, except for an unexpected increase in the later growth stage in 2014 (Figure 4). Specifically, the maximum values of the water indices (WI, NDWI, and NDII) occurred at 59–70 DAS, while the maximum values of the structural indices (NDVI and OSAVI) occurred at 52–59 DAS in the T1–T5 treatments in 2013. In comparison to the treatments in 2013, those in 2014 showed few differences in the dynamics of vegetation, with an increasing trend after 72 DAS. In addition, water stress exerted significant negative effects on the SVIs during the two growing seasons.

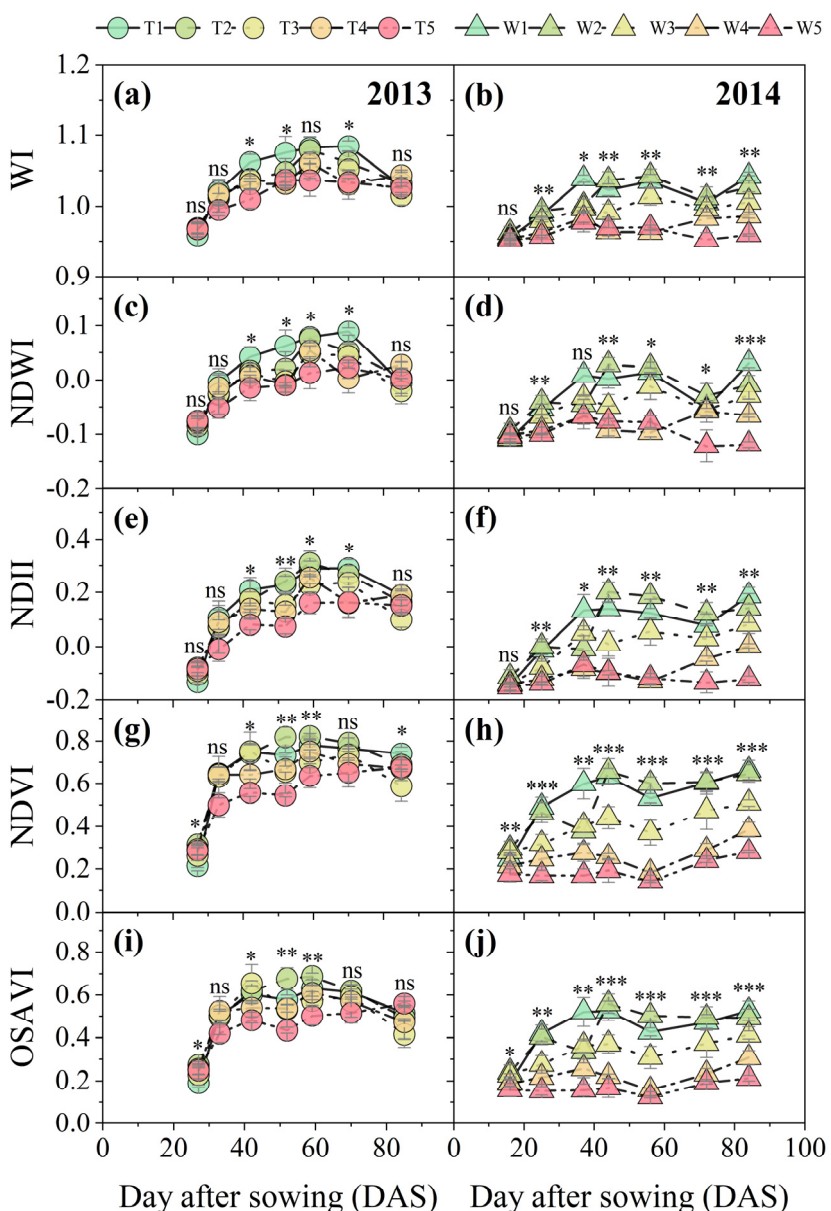

**Figure 4.** Temporal variability in vegetation indices (WI, NDWI, NDII, NDVI, and OSAVI) during soil drying among different irrigation treatments in the 2013 (**a,c,e,g,i**) and 2014 (**b,d,f,h,j**) growing seasons. WI, water index; NDWI, normalized difference water index; NDII, normalized difference infrared index; NDVI, normalized difference vegetation index; and OSAVI, optimal soil-adjusted vegetation index. The "***", "**", and "*" denote significant differences among treatments based on the analysis of variance (ANOVA) at $p < 0.001$, $p < 0.01$, and $p < 0.05$ levels, respectively, while "ns" indicates no significant differences. Error bars denote mean $\pm$ SE.

*3.4. Spectral Vegetation Index Responses to CWC and VWC Variations during Soil Drying in the Growing Season*

Both the water-related indices (WI, NDWI, and NDII) and structure-related indices (NDVI and OSAVI) shared a significant linear relationship with CWC, while they exhibited a nonlinear relationship with VWC (Figure 5). Furthermore, in comparison to NDVI ($R^2 = 0.69$, $p < 0.001$, root mean square error (RMSE) = 0.116) and OSAVI ($R^2 = 0.64$, $p < 0.001$, RMSE = 0.097), the WI ($R^2 = 0.75$, $p < 0.001$, RMSE = 0.019), NDWI ($R^2 = 0.71$, $p < 0.001$, RMSE = 0.029) and NDII ($R^2 = 0.79$, $p < 0.001$, RMSE = 0.063) showed a closer connection with CWC for the LAI values, as the values of $R^2$ (R-squared) and RMSE suggested.

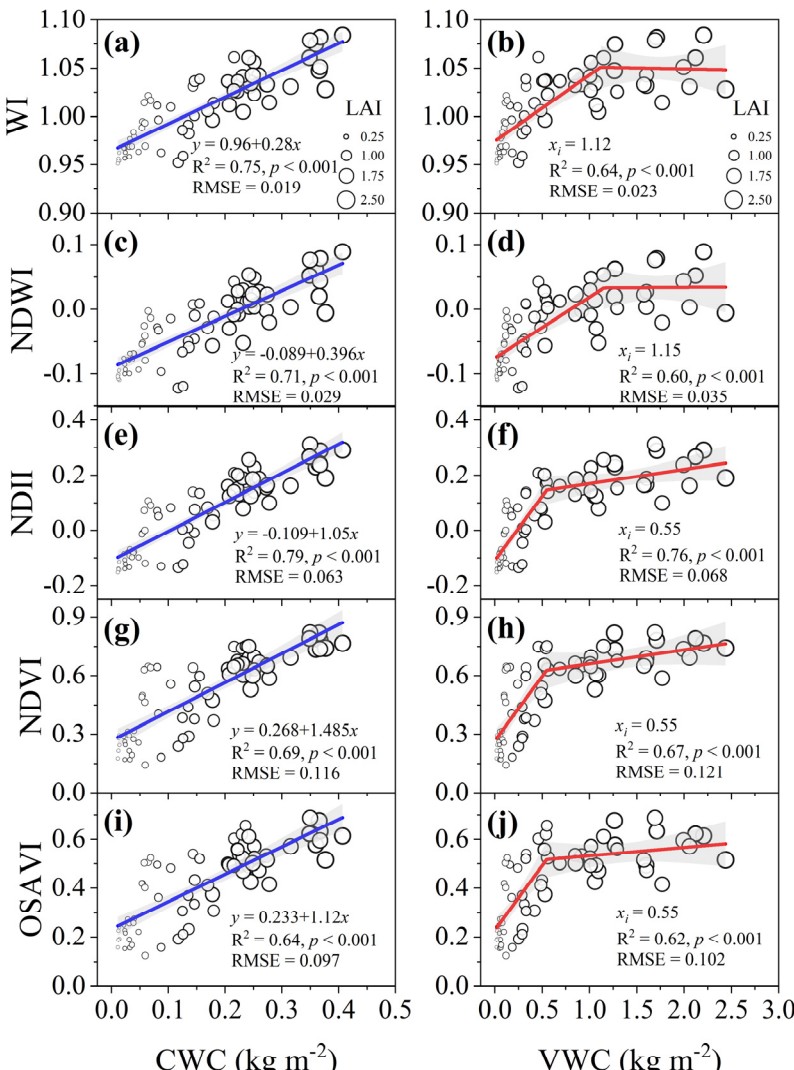

**Figure 5.** Response of the spectral vegetation indices to canopy water content (CWC) and vegetation water content (VWC) during soil drying. (**a**) WI, (**c**) NDWI (**e**) NDII, (**g**) NDVI, and (**i**) OSAVI response to CWC, respectively. (**b**) WI, (**d**) NDWI, (**f**) NDII, (**h**) NDVI, and (**j**) OSAVI response to VWC, respectively. Linear and nonlinear regression lines show the 95% confidence intervals. RMSE, root mean square error.

Moreover, the critical values where the SVIs responded to VWC were 1.12 kg m$^{-2}$ and 1.15 kg m$^{-2}$ for the WI and NDWI (Figure 5b,d), respectively, while the NDII, NDVI, and OSAVI had the same critical value of 0.55 kg m$^{-2}$ (Figure 5f,h,j). The NDII, NDVI, and OSAVI tended to be easier to saturate at low LAI values than the WI and NDWI, as indicated by the LAI to some degree. The results indicated that, in comparison to the NDVI and OSAVI, the WI, NDWI, and NDII were better able to capture CWC variations during soil drying. However, all five SVIs captured the VWC variations before a threshold value of VWC, but they did not sensitively reflect the VWC variations after that point, as the values of the SVIs did not increase significantly with the increase in VWC (Figure 5b,d,f,h,j).

### 3.5. Linking the Dynamic Variations in SVIs with Abiotic and Biotic Factors

The correlation analysis suggested that ASWC had significantly negative correlations ($p < 0.05$) with LAI, CWC, VDM, and VWC, while the ASWC did not show any significant correlation with the SVIs (Figure 6). In contrast, variations in LAI, CWC, VDM, and VWC exhibited significant relationships with the SVIs during soil drying. In addition, the correlation coefficient for each pair of SVIs was greater than 0.89.

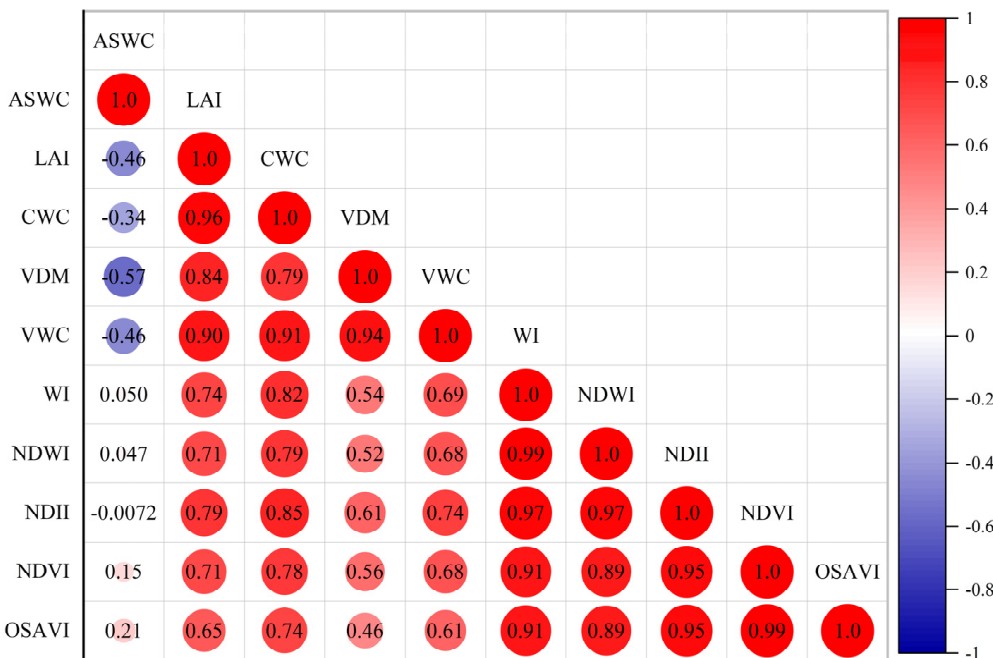

**Figure 6.** A visualization of Pearson correlation matrix of canopy characteristics and the spectral vegetation indices. ASWC, available soil water content; LAI, leaf area index; CWC, canopy water content; VDM, vegetation dry mass; VWC, vegetation water content; WI, water index; NDWI, normalized difference water index; NDII, normalized difference infrared index; NDVI, normalized difference vegetation index; and OSAVI, optimal soil-adjusted vegetation index.

Moreover, SEM analysis revealed that the combination of these direct and indirect effects via the five key variables accounted for 79.3%, 74.1%, 81.8%, 82.2%, and 78.8% of the variances in WI, NDWI, NDII, NDVI, and OSAVI, respectively (Figure 7). The dynamics of the WI, NDWI, NDII, and OSAVI were directly regulated by CWC and ASWC, and indirectly modulated by LAI, VWC, and ASWC (Figure 7a–c,e). In contrast, the NDVI was directly influenced by CWC, ASWC, and VDM but was indirectly controlled by LAI and VWC (Figure 7d).

Based on the standardized total effects from the SEM, the variations in the WI, NDWI, NDII, and OSAVI were mainly regulated by CWC, LAI, and VWC, while those in the NDVI were mainly regulated by CWC, LAI, and VDM (Figure 8). ASWC affected OSAVI (standardized total effects = 0.209) and NDVI (standardized total effects = 0.147) more than the other three SVIs (standardized total effects < 0.05). Overall, CWC was the most important direct biotic driver of SVIs during soil drying over the entire growing season, while LAI was the most important indirect biotic driver. Additionally, VWC was a critical indirect regulator of the dynamics of the WI, NDWI, NDII, and OSAVI, whereas VDM was a critical indirect regulator of the dynamics of the NDVI.

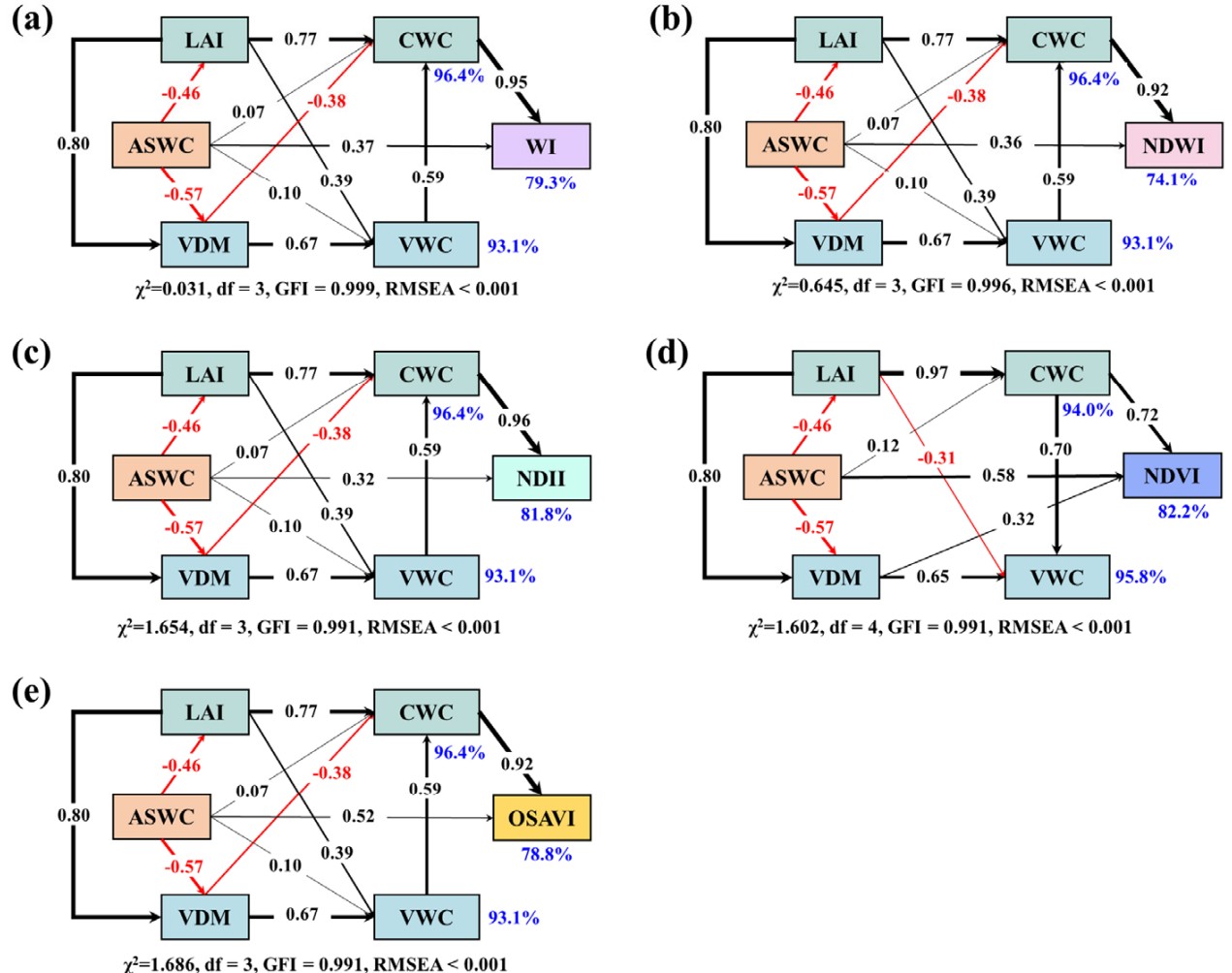

**Figure 7.** Pathways for abiotic and biotic factors influencing spectral vegetation index during soil drying in the growing season. Structural equation model (SEM) analyses were conducted for five spectral vegetation indices: (**a**) WI, (**b**) NDWI, (**c**) NDII, (**d**) NDVI, and (**e**) OSAVI. Numbers on arrows are standardized path coefficients, and black or red arrows indicate positive or negative relationships between variables, respectively. Goodness-of-fit statistics for the models are shown below the model. ASWC, available soil water content; LAI, leaf area index; CWC, canopy water content; VDM, vegetation dry mass; VWC, vegetation water content in all leaves and stems; WI, water index; NDWI, normalized difference water index; NDII, normalized difference infrared index; NDVI, normalized difference vegetation index; and OSAVI, optimal soil-adjusted vegetation index. $\chi^2$, chi-squared; df, degree of freedom; GFI, goodness-of-fit index; RMSEA, root mean square error of approximation.

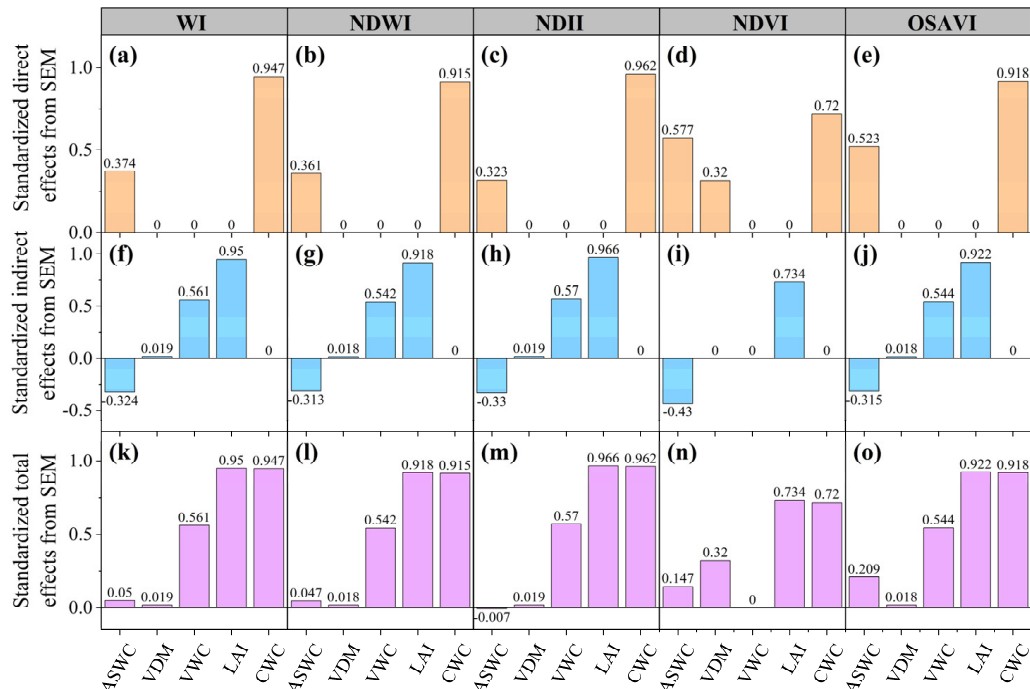

**Figure 8.** Standardized direct (**a–e**), indirect (**f–j**), and total effects (**k–o**) on the spectral vegetation indices derived from the structural equation model (SEM). ASWC, available soil water content; LAI, leaf area index; CWC, canopy water content; VDM, vegetation dry mass; VWC, vegetation water content; WI, water index; NDWI, normalized difference water index; NDII, normalized difference infrared index; NDVI, normalized difference vegetation index; and OSAVI, optimal soil-adjusted vegetation index.

## 4. Discussion

### 4.1. Spectral Vegetation Index Responses to CWC and VWC Variations during Soil Drying in the Growing Season

Instantaneous water stress and long-term water stress are two kinds of plant water stress at different phases [34,53]. The impact of instantaneous water stress can be reflected in the decrease in leaf water content, which influences stomatal activity and gaseous exchange between plants and the ambient atmosphere [35,54,55]. However, long-term water stress can significantly affect crop structural characteristics such as LAI, aboveground biomass, and grain yield [37,39]. Thus, if instantaneous water stress is not adequately detected and managed, it creates the basis for long-term water stress that contributes to extensive crop yield loss [35]. Fortunately, remote sensing technology can be effectively used to monitor plant water stress. Previous studies demonstrated that SVIs based on remote sensing data collected at a specific growth stage can retrieve plant water content at the leaf and canopy levels [16,54,56,57]. However, little attention has been given to the dynamic variations in plant water status during soil drying.

Our results showed that the dynamic characteristics of CWC and VWC shared a similar dynamic trend during the growing season and that impacts of different irrigation treatments on plant water status were obvious (Figure 1a,b,d,e; Table S1). Besides this, the ratio of CWC to VWC displayed a declining trend (Figure 1c,d). In other words, the water taken in by the stems increased with the growth stage while that taken in by the leaves decreased throughout the growing seasons. This finding suggested that water storage in stems might play an important role in regulating the plant water status, as suggested by [29,58]. Moreover, all five SVIs showed an obvious dynamic trend of first increasing and then decreasing (Figure 4), which was also reported in another study [59]. In addition, the spectral vegetation index values indicated the significantly negative impacts of soil drying on crop reflectance among the treatments during the growing seasons. The results

confirmed that spectral vegetation index values clearly varied during soil drying in the growing season. Therefore, the dynamics of plant water status need to be considered in when assessing the effects of drought on crop production.

### 4.2. Relationships between CWC, VWC, and the SVIs

Vegetation water status plays a key role in plant functioning, and water and energy exchange with the atmosphere, especially during a drought [8]. Vegetation water status also has been estimated in many studies with various definitions at both leaf and canopy scales [9,12,13,19,30,60], which makes it difficult to compare. For a variety of vegetation types, the largest unknown for predicting vegetation water status is stem water content, which is assumed to be allometrically related to the plant canopy water content [20]. To address the importance of stem water content, in this study, we used CWC to refer to the total water mass of all plant leaves per unit of ground area, and VWC to represent the total amount of water in the stems and leaves per unit of ground area. A previous study found that CWC shared a significantly linear relationship with VWC based on in situ measurements obtained from a series of maize field experiments [12]. By comparison, our results found an apparent nonlinear relationship between CWC and VWC (Figure 2), which likely occurred due to the differences in the measurement durations during field experiments and crop exposure to soil drying. Moreover, although different irrigation amounts were applied at two growth stages in the two growing seasons, the nonlinear relationships between CWC and VWC were generally consistent (Figure 2).

In addition, many SVIs have been used to estimate plant water stress [23,61]. In general, SVIs related to plant water stress can be divided into three groups: xanthophyll indices (e.g., photochemical reflectance index; PRI), structural indices (e.g., NDVI and OSAVI), and water indices (e.g., WI, NDWI, and NDII). Some studies have reported that structural vegetation indices (e.g., NDVI and OSAVI) tend to correlate with canopy water status [30], but they are not always good indicators of plant water stress as they are directly related to canopy water absorption [13,15,31]. Our results showed that CWC was better represented by the WI, NDWI, and NDII than the NDVI and OSAVI (Figure 5). Specifically, in comparison to the NDVI, the NDII exhibits a better linear relationship with CWC [62], which was also confirmed by our results (Figure 5). In addition, the NDII is allometrically related to VWC [21], which was in agreement with our result (Figure 5). Moreover, in comparison to the water indices (WI, NDWI, and NDII), the structural indices (NDVI and OSAVI) were likely determined more easily, because NDVI and OSAVI are sensitive to both LAI and leaf water content [20]. Furthermore, the NDVI and OSAVI are based on the red and near-infrared bands, which are in the strong chlorophyll absorption range and high reflectance plateau of the vegetation canopy, respectively. The NDVI and OSAVI account for differences in LAI and canopy structure, but the chlorophyll extinction coefficients are very large at red wavelengths; thus, NDVI and OSAVI saturates at a lower LAI compared to NDII [20]. Therefore, estimation of plant water stress at the canopy scale will differ depending on whether it is water in the stems or in other parts of a plant (CWC vs. VWC) being considered.

### 4.3. Impact Mechanism of the Soil Drying Process on the SVIs

Soil moisture deficit may be the most widely accepted indicator for assessing agricultural drought. Thus, soil moisture mapping is an effective way to identify potential crop drought, although, there are still many technical barriers to directly measuring soil moisture across the effective root zone (approximately 30–50 cm) of grain crops through remote sensing [63,64]. However, soil moisture may not be the core issue when evaluating drought impacts on agriculture, as a water deficit does not necessarily result in biomass or yield loss [42,65,66]. In fact, in comparison to soil moisture, plant water status is more related to plant physiological functions (e.g., photosynthesis, transpiration, and respiration) [29,67] and is directly connected to biomass and yield production. In addition, leaf water content within a plant canopy plays an important role in light penetration and scattering [68].

Variations in plant water status can be captured by the differences in canopy reflectance information. Thus, the remotely sensed estimation of plant water status using optical data is feasible and useful for monitoring and assessing for agricultural drought [34]. Consequently, it is necessary to understand the underlying mechanism of spectral vegetation index responses to the impact of water stress on the dynamics of plant water content.

Our results based on SEM analysis suggested that the combination of these direct and indirect effects via five key canopy variables can explain 74.1~82.2% of the variances in the five SVIs (Figure 7). Specifically, of the variables, CWC was the most important direct biotic driver of the five SVIs during soil drying in the growing season, while LAI was the most important indirect biotic driver. As suggested by previous studies, LAI is functionally related to the exchange of water/energy between vegetation and the atmosphere [69,70] and plays an important mediating role in the soil–plant–atmosphere continuum [71]. In addition, ASWC affected the NDVI (standardized total effects = 0.147) and OSAVI (standardized total effects = 0.209) more than the other three SVIs (standardized total effects < 0.05) (Figure 8), as soil noise can significantly influence the ability of the NDVI and OSAVI to detect sparse vegetation [72,73]. Furthermore, VWC was a critical indirect regulator of WI, NDWI, NDII, and OSAVI dynamics, whereas VDM was a critical indirect regulator of the NDVI. That is, the water content and biomass characteristics of stems played an important indirect role in adjusting the plant water status. Nevertheless, many maize varieties and crop species need to be considered for more years and at large spatial scales of observation to further improve our understanding of this issue in the future.

## 5. Conclusions

This study evaluated the capability of SVIs to reflect the dynamic variations in CWC and VWC, and explored the underlying mechanisms of these variations during soil drying, based on a two-year field experiment. The results suggested that CWC, VWC, and the SVIs had obvious dynamic trends of first increasing and then decreasing during the entire growing season. Soil drying had significant impacts on CWC, VWC, and the SVIs in the different irrigation treatments. VWC was allometrically related to CWC under different irrigation amounts across growing seasons. The five SVIs showed a good linear relationship with CWC, and the indices had a nonlinear relationship with VWC. Furthermore, the critical values of the SVIs when applied to retrieve VWC were 1.12 kg m$^{-2}$ and 1.15 kg m$^{-2}$ for the WI and NDWI, respectively, while those of the NDII, NDVI, and OSAVI were the same at 0.55 kg m$^{-2}$. Consequently, the WI and NDWI were superior to the NDII, NDVI, and OSAVI in reflecting plant water content given their comprehensive sensitives to CWC and VWC. Moreover, SEM analysis suggested that of the variables, CWC was the most important direct biotic driver of the SVIs during soil drying, while LAI was the most important indirect biotic driver. Additionally, VWC was a critical indirect regulator of the WI, NDWI, NDII, and OSAVI dynamics, whereas VDM was a critical indirect regulator of the NDVI dynamics. Overall, the results of this study may provide additional information for estimating plant water content using spectral data and insights into the impact of the soil water deficit mechanism on SVIs.

**Supplementary Materials:** The following are available online at https://www.mdpi.com/article/10.3390/rs14030584/s1, Figure S1: time series of air temperature (a,b), precipitation (c,d), photosynthetically active radiation (e,f), and relative humidity (g,h) during the 2013–2014 growing seasons. Table S1: The mean values of LAI, CWC, VWC and spectral vegetation indices during soil drying under different irrigation treatments in the 2013 and 2014 growing seasons.

**Author Contributions:** Conceptualization, H.Z., G.Z. and Q.H.; methodology, H.Z., G.Z. and Q.H.; validation, H.Z., G.Z., Q.H. and X.S.; formal analysis, H.Z.; investigation, H.Z. and X.S.; writing—original draft preparation, H.Z. and G.Z.; writing—review and editing, H.Z., G.Z. and Q.H.; funding acquisition, G.Z. All authors have read and agreed to the published version of the manuscript.

**Funding:** This research was funded by National Natural Science Foundation of China (42130514, 42141007) and Basic Research Fund of Chinese Academy of Meteorological Sciences (2020Z004).

**Institutional Review Board Statement:** Not applicable.

**Informed Consent Statement:** Not applicable.

**Data Availability Statement:** Data available on request.

**Acknowledgments:** We thank Xueyan Ma, Yaohui Shi, Qiuling Wang, Minzheng Wang, Xiaoyu Feng, and Fan Wang from the Chinese Academy of Meteorological Sciences for their work assistance, both in the field and in laboratory analysis. We also thank Feng Zhang at the Institute of Botany, CAS for technical assistance in hyperspectral measurement. Sincere thanks also go to the editor and anonymous reviewers for their thoughtful comments that improved this manuscript.

**Conflicts of Interest:** The authors declare no conflict of interest.

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
