# Peer review of "Dynamic Characteristics of Canopy and Vegetation Water Content during an Entire Maize Growing Season in Relation to Spectral-Based Indices"

_remotesensing, doi:10.3390/rs14030584_

Round 1
Reviewer 1 Report
Dear authors, congratulations on your work. Please, kindly find some comments below.
Please verify the following expressions:
line 225: Correction analysis or correlation analysis?
line 255: Significant correction or significant correlation?
line 306: Correction analysis or correlation analysis?
line 310: Negative corrections or negative correlations?
line 310: Correction coefficient or correlation coefficient?
Figures 7a and 7b: CDM or VDM?
Figure 8: LWC or CWC?
Please, what do you mean by "combination of these direct and indirect effects" in line 317? How do they explain the ranging from 74.1% to 82.2% of the variances in the spectral vegetation indices? Please, explain better such a dynamic when observing Figure 7.
Please, give the name or meaning to the variables when employing them for the first time in the text. For example: p, RMSE, R, GFI, RMSEA.
Reviewer 2 Report
The manuscript " Dynamic Characteristics of Canopy and Vegetation Water Content during an Entire Maize Growing Season in Relation to Spectral-based Indices” authored by Zhou et al. employ well-proven methodologies and present the results in a convincing manner. However, a number of minor improvements are required before accepting this paper.
- Line 3: The first letter of during, “d” should be in the same font as others in the title.
- The abstract has 338 words, but according to the journal, the abstract should be a total of about 300 words maximum.
- Line 214: “2014were” ---> “2014 were”
- Line 391: “referred” ---> “refer”
- Line 455: “on” ---> “of”
- Line 479: “photosynthetically available radiation” ---> “photosynthetically active radiation”. Also, need to correct in the supplementary materials.
Reviewer 3 Report
Dears authors,
I have read your article carefully. Methodology used is adequate for the proposed objectives. Authors have carried out a study in which they correlate canopy water content (CWC) and complete plant water content (VWC) with different spectral indices.
References are numerous, suitables and very currents. Content and extension of the abstract are suitable and give a clear idea of ​​the paper. The work is well focused, It is a very complete study and english language is correct.
Manuscript contains valuable and interesting information that deserve to be published, but authors not show the data. I think that for this article to be published, the authors must include a table with the data (mean ± SD) of the variables they have determined (CWC, VWC, LAI, WI, NDWI, NDII, NDVI and OSAVI). Authors give little information about the soil used.
Highlights must be modified according to the notes I have put in the attached pdf.
I attach a pdf file where the authors will be able to find observations and comments that should be considered and that might improve this manuscript.

Reviewer 4 Report
Article is very good and interesting
Author Response
Please see the attachment

This manuscript is a resubmission of an earlier submission. The following is a list of the peer review reports and author responses from that submission.